# Biodentine^™^ Boosts, WhiteProRoot^®^MTA Increases and Life^®^ Suppresses Odontoblast Activity

**DOI:** 10.3390/ma12071184

**Published:** 2019-04-11

**Authors:** Anabela Paula, Mafalda Laranjo, Carlos Miguel Marto, Ana Margarida Abrantes, João Casalta-Lopes, Ana Cristina Gonçalves, Ana Bela Sarmento-Ribeiro, Manuel M. Ferreira, Maria Filomena Botelho, Eunice Carrilho

**Affiliations:** 1Institute of Integrated Clinical Practice, Institute for Clinical and Biomedical Research (iCBR), Area of Environment Genetics and Oncobiology (CIMAGO), CNC.IBILI, Faculty of Medicine, University of Coimbra, 3000-548 Coimbra, Portugal; eunicecarrilho@gmail.com; 2Biophysics Institute, Institute for Clinical and Biomedical Research (iCBR), Area of Environment Genetics and Oncobiology (CIMAGO), CNC.IBILI Consortium, Faculty of Medicine, University of Coimbra, 3000-548 Coimbra, Portugal; mafaldalaranjo@gmail.com (M.L.); mabrantes@fmed.uc.pt (A.M.A.); joao.casalta@gmail.com (J.C.-L.); mfbotelho@fmed.uc.pt (M.F.B.); 3Institute Experimental Pathology, Institute for Clinical and Biomedical Research (iCBR), Area of Environment Genetics and Oncobiology (CIMAGO), CNC.IBILI, Faculty of Medicine, University of Coimbra, 3000-548 Coimbra, Portugal; mig-marto@hotmail.com; 4Radiation Oncology Department, Coimbra University Hospital Center, 3000-548 Coimbra, Portugal; 5Laboratory of Oncobiology and Hematology, Institute for Clinical and Biomedical Research (iCBR), Area of Environment Genetics and Oncobiology (CIMAGO), CNC.IBILI Consortium, Faculty of Medicine, University of Coimbra, 3000-548 Coimbra, Portugal; acc.goncalves@gmail.com (A.C.G.); absarmento@fmed.uc.pt (A.B.S.-R.); 6Institute of Endodontics, Institute for Clinical and Biomedical Research (iCBR), Area of Environment Genetics and Oncobiology (CIMAGO), CNC.IBILI, Faculty of Medicine, University of Coimbra, 3000-548 Coimbra, Portugal; m.mferreira@netcabo.pt

**Keywords:** biocompatibility, biomaterials, cytotoxicity, dentinogenesis, pulp capping, odontoblast

## Abstract

(1) Background: When pulp exposure occurs, reparative dentinogenesis can be induced by direct pulp capping to maintain the vitality and function of the tissue. The aim of this work was to assess the cytotoxicity and bioactivity of three different direct pulp capping materials, calcium hydroxide (Life^®^), mineral trioxide aggregate (WhiteProRoot^®^MTA) and calcium silicate (Biodentine^™^), in an odontoblast-like mouse cell line (MDPC-23). (2) Methods: Metabolic activity was assessed by the 3-(4,5-dimethylthiazol-2-yl)-2,5-diphenyltetrazolium bromide test (MTT)assay, viability by the sulforhodamine B (SRB) assay, and the type of death and cell cycle analysis by flow cytometry. Alkaline phosphatase was evaluated by polymerase chain reaction (PCR), and dentin sialoprotein expression was assessed by immunocytochemistry. Mineralization was determined by the Alizarin Red S colorimetric assay and quantified by spectrophotometry. (3) Results: Life^®^ induced a decrease in metabolic activity and viability, which is associated with an increase cell death. WhiteProRoot^®^MTA and Biodentine™ induced similar effects in cytotoxicity assays, with an increase in the expression of dentin sialoprotein (DSP) and formation of mineralized deposits, especially with Biodentine™. (4) Conclusions: The results of WhiteProRoot^®^MTA confirm its indication for these therapies, justifying its recognition as the “gold standard”. Biodentine™ may be an alternative, since they promote the same cellular response that mineral trioxide aggregate (MTA) does.

## 1. Introduction

Clinical situations as deep cavities, severe crown trauma, and iatrogenic circumstances may expose the pulp tissue to the external oral environment. Dental pulp tissue is potentially able to repair itself through dentinogenesis. During this process, the damaged odontoblasts are replaced by newly differentiated odontoblast cells [1,2]. Migration of progenitor cells to the lesion site and subsequent proliferation and differentiation of these cells into odontoblasts occurs [3,4,5]. Thus, when a dental pulp tissue exposition occurs, reparative dentinogenesis can be induced by pulp capping to maintain the vitality and function of the pulp. There is a consensus that treatment of pulp exposure requires the use of materials that promote pulp tissue repair and the formation of mineralized tissue between the pulp tissue and the repair material [6,7,8,9].

In the past, calcium hydroxide cement was used among gold standard therapies, due to its important physicochemical characteristics for the treatment success. This biomaterial is the most studied and documented in several cellular, animal and clinical studies and presents satisfactory results with success rates up to 80%. However, there are some disadvantages such as poor adhesion to dentin, high solubility and mechanical instability, and, consequently, dentin-bridged material with multiple tunnel defects [2,10,11].

Developed in the 1990s, mineral trioxide-based cement received the highest attention, initially as a retrograde obturation material. Its characteristics expanded the indications, being used as material for direct pulp capping. The mineral trioxide aggregates promote the proliferation and differentiation of pulp cells and demonstrate the formation of mineralized tissue during the development of the dentin bridge after pulp exposure. In contrast to calcium hydroxide-based materials, the mineral trioxide aggregates induce less inflammation of the pulp tissue and limit cell necrosis [12,13,14,15].

Recently, tricalcium silicate cements have emerged. This biomaterial is calcium silicate-based and has a similar composition to mineral aggregate trioxide cements, but with substantially better setting times due to calcium chloride in addition to other clinical advantages. This material is characterized by its dissolution in calcium hydroxide and, when in contact with the tissues, leads to the formation of hydroxyapatite [16,17,18,19,20,21]. Considering the enhanced characteristics of this compound, we hypothesized that tricalcium silicate cements, such as Biodentine^™^, may have a lower cytotoxicity and better bioactivity than the available materials for pulp capping therapy.

Therefore, the objective of this study was to evaluate the response of Mouse Dental Papilla Cell-23 line (MDPC-23), odontoblast-like cells, submitted to three biomaterials, including the gold standard mineral trioxide aggregate. The importance of cellular response justifies the need to evaluate cytotoxicity and bioactivity in terms of underlying cellular mechanisms, focusing on the reparative dentinogenesis, with the assessment of the differentiation and mineralization processes.

## 2. Materials and Methods

### 2.1. Materials

Three commercially available biomaterials were used in the present study: Life^®^ (Kerr Hawe, Bioggio, Switzerland) as a calcium hydroxide-based cement, WhitePro^®^Root MTA (Dentsply, Maillefer, Tulsa Dental Specialties, Switzerland) as an aggregated trioxide mineral-based cement, and Biodentine^™^ (Septodont, Saint-Maur-des-Fosses, France) as a calcium silicate-based cement.

### 2.2. Sample Preparation

Life^®^, WhitePro^®^Root MTA and Biodentine^™^ were mixed according to the manufacturers’ instructions and shaped with 1.5-mm-thick polyvinyl chloride molds with a diameter of 3 mm, under aseptic conditions. Life^®^ and Biodentine^™^ were kept for 24 h at 37 °C at 95% relative humidity atmosphere while the WhiteProRoot^®^MTA was kept for 72 h under the same conditions in a HeraCell incubator 150. After setting, the disks were exposed to ultraviolet light for 20 min on each surface to ensure sterility. Material-conditioned media were prepared in accordance with ISO standard 10993-5. Thus, the cell culture media were incubated with the substrate in a ratio of 250 mm^2^ on the surface of the material per mL of culture medium for 24 h at 37 °C [22,23]. The conditioned media were diluted to obtain five serial dilutions: 100%, 50%, 25%, 12.5%, and 6.25%. Dulbecco’s Modified Eagle’s Medium (DMEM) (Sigma D-5648, Sigma, Kawasaki, Japan) not submitted to the materials, incubated for 24 h at 37 °C, was used as a control. The pH of the conditioned media was evaluated using samples of each concentration. No pH adjustment was performed [22].

### 2.3. Cell Culture

The spontaneously immortalized MDPC-23 cell line, provided by Professor Jacques Nör (University of Michigan, Ann Arbor, MI, USA), was cultured using DMEM culture medium supplemented with 10% FBS (Sigma F7524), 250 μM sodium pyruvate (Gibco 11360) and 1% of antibiotic (100 U/mL penicillin and 10 μg/mL streptomycin; Sigma A5955). The cell line was maintained at 37 °C in a humidified atmosphere with 95% air and 5% CO_2_ in a HeraCell 150 incubator.

### 2.4. Metabolic Activity

To evaluate the effect of the biomaterials on the cell’s metabolic activity, the MTT test (3-(4,5-dimethylthiazol-2-yl)-2,5-diphenyltetrazolium bromide) (Sigma M2128) was performed. Cell cultures were incubated with MTT (0.5 mg/mL) overnight in the dark at 37 °C. To solubilize the obtained formazan crystals, a 0.04 M solution of hydrochloric acid in isopropanol was added to each well and the plates were allowed to stir for 30 min. The absorbance was quantified at 570 nm with a reference filter of 620 nm using ELISA (Synergy HT spectrophotometer, Biotek^®^, Bad Friedrichshall, Germany) [24]. The metabolic activity was expressed as a percentage of control cell cultures.

### 2.5. Cell Viability

The sulforhodamine B (SRB) assay allows assessment of cell viability by correlation with total protein content. Briefly, the cell cultures were fixed and dyed with a 4% concentration solution of sulforhodamine B (Sigma S9012) in 1% strength acetic acid for 2 h in the dark [25]. After washing and drying, they were solubilized with tris-NaOH (10 nm) (Sigma T1503 and Sigma S5881). Absorbance reading was done at 540 nm, with a reference filter of 690 nm, on ELISA (Synergy HT spectrophotometer, Biotek^®^, Bad Friedrichshall, Germany). The results were expressed as a percentage of control cell cultures.

### 2.6. Cell Proliferation

Cell count was performed on Hemascreen 18 (Hospitex Diagnostics, Florence, Italy). The expressed results translate the ratio between the number of cells counted and the number of plated cells.

### 2.7. Cell Cycle

Cell cycle was evaluated by flow cytometry using propidium iodide PI/RNase solution (Immunostep PI/RNase) was used according to manufacturer’s instructions. The evaluation was performed on the FACSCalibur cytometer (BD Biosciences, Qume Drive, San Jose, CA, USA) and quantification of the information was performed using specific software (Paint-a-Gate 3.02, Macintosh Software). The results were presented as percentage of cells in the Pre-G0 phase, the G0/G1 phase, the S phase, or the G2/M phase.

### 2.8. Types of Cell Death

To evaluate the types of cell death, double labeling with annexin V (AnV) and propidium iodide (PI) was performed according to the manufacturer’s instructions (Immunostep ANXVFKIT Immunotech and KIT Immunotech) [26]. The analysis was performed on the FACSCalibur cytometer (BD Biosciences, Qume Drive, San Jose, CA, USA). The results are presented as a percentage of living cells, cells in apoptosis, cells in late apoptosis/necrosis, and cells in necrosis.

### 2.9. Mitochondrial Membrane Potential

The mitochondrial membrane potential was determined using a fluorescent probe, JC-1 (5,5′,6,6′-tetrachloro-1,18,3,3-tetraethylbenzimidazolcarbocyanine iodide) (Sigma T4069). Cells were labelled as in Laranjo et al., 2013 [26]. Detection was performed on the FACSCalibur cytometer (BD Biosciences, Qume Drive, San Jose, CA, USA). The results are presented correspond to the ratio of the mean fluorescence intensity (MIF) for the monomers and for the aggregates.

### 2.10. Cellular Morphology

Morphological characteristics were evaluated by optical microscopy (Nikon Eclipse NI optical microscope, Nikon Instruments, Amsterdam, Netherlands) after staining with May-Grünwald Giemsa medium (Sigma 32856). The photographs were obtained from a Nikon OS-Fi2 camera (Nikon Metrology Inc, Irvine, CA, USA) and the images were subsequently analyzed using the NIS-Elements D software (version 4.00). Photographs with magnifications of 100×, 200× and 500× were obtained.

### 2.11. Oxidative Stress

Intracellular expression of peroxides was determined by flow cytometry through intracellular oxidation of the non-fluorescent probe DCFH2-DA (2′,7′-dichlorodihydrofluorescein diacetate) (Molecular probes, Invitrogen). The cells cultures were incubated for 45 min, in the dark, at 37 °C with 5 μM DCFH2-DA (Molecular probes, Invitrogen). Evaluation was performed on the FACSCalibur cytometer (BD Biosciences, Qume Drive, San Jose, CA, USA). The results are presented as the MIF percentage relative to the control cell cultures.

### 2.12. Alkaline Phosphatase—Gene Expression

Quantitation of alkaline phosphatase (ALP) gene expression by real-time reverse transcription polymerase chain reaction (RT-PCR) was evaluated by relative quantification using the BioRad SybrGreen amplification detection system using the ΔΔCt method [27]. RNA was extracted from the cells using NZYol (NZYtech) and converted into complementary DNA using a Superscript III kit (Invitrogen). The cDNA of each of the samples was amplified and quantified by the use of forward and reverse primers (*Gapdh*: forward 5′-GACAACTTTGGCATCGTGGA-3′ and reverse 5′-ATGCAGGGATGATGTTCTGG-3′; *Alpl*: forward 5′-CGCCTATCAGCTAATGCACAACA-3′ and reverse 5′-ATGAGGTCCAGGCCATCCAG-3′) [28,29,30]. The results represent the mean ± standard error of four independent experiments and are expressed in variation of the Gapdh gene.

### 2.13. Expression of Dentin Sialoprotein (DSP)

The dentin sialoprotein was evaluated through immunocytochemistry. Cells were prepared and incubated with a primary antibody (DSP (M-20) Antibody, Santa Cruz Biotechnology, Europe, 1:100) and a peroxidase-conjugated secondary antibody (Polyclonal Rabbit Anti-Goat immunoglobulins/HRP, Dako, Denmark, 1:100). Images were obtained on a Moticam 1080 optical microscope (Microscope Central, Bustleton Pike, Feasterville, PA, USA).

### 2.14. Mineralized Nodule Formation

To determine the degree of mineralization, the cell cultures were stained with a solution of alizarin red S (3,4-dihydroxy-9,10-dioxo-2-anthracenesulfonic acid sodium salt) (A5533 Sigma-Aldrich). Cell cultures were fixed with 4% paraformaldehyde for 15 min and stained with a solution of Alizarin Red Staining (40 mM, pH 4.2) [31,32,33,34,35] for 20 min at 37 °C [36,37] and were photographed using a MoticamPro 285A optical microscope (Microscope Central, Bustleton Pike, Feasterville, PA, USA). Complementarily, extraction was performed using a solution of 10% (w/v) acetic acid and 20% (w/v) methanol. The absorbance of extracts with 490 nm was measured on a FACSCalibur cytometer (BD Biosciences, Qume Drive, San Jose, CA, USA) [32]. The results of the quantitative analysis were presented as the mean ± standard error in relation to the control.

### 2.15. Statistical Analysis

Statistical analysis was performed using IBM^®^ SPSS^®^ software version 20. The Shapiro-Wilk test was used to evaluate the normal distribution of the quantitative variables. One-way analysis of variance (ANOVA) test was performed when a normal distribution and homogeneity of variances were verified. When the contrary was found, the Kruskal-Wallis test was performed. Multiple comparisons were then made between experimental groups using the Student’s t-test, and non-parametric tests such as the Kruskal-Wallis test or the Mann-Whitney-Wilcoxon test were used to test the heterogeneity of two ordinal samples. All multiple comparisons were corrected according to the Bonferroni method and a significance value of 5% was considered for all comparisons. The results were expressed as mean ± SE.

## 3. Results

### 3.1. Biomaterials Extracts pH

The pH of the conditioned media was in the alkaline range spectrum. The maximum value was of 10.90 ± 1.30 for the WhiteProRoot^®^MTA at 100% concentration while the minimum value was 7.93 ± 0.31 for Biodentine^™^ at the concentration of 6.25%. For each biomaterial, the pH decreases as the concentration of conditioned medium decreases, approaching the lower concentrations near neutral pH.

### 3.2. Metabolic Activity and Cell Viability

The experimental results of metabolic activity and cell viability of the MDPC-23 cell line are shown in Figure 1.

Treatment with Life^®^ determined a significant decrease in metabolic activity at concentrations equal to and greater than 12.5% of conditioned media. The cell viability was significantly decreased in treated cell cultures. Treatment with Biodentine^TM^ also determined a statistically significant decrease (*p* < 0.001) of the metabolic activity when the cells were incubated with the highest concentration studied, as well as in cell viability at 100% and 50% concentrations at all times. In contrast, WhiteProRoot^®^MTA did not determine significant changes in cellular metabolic activity or cellular viability.

### 3.3. Cell Proliferation

The proliferation of cell cultures treated with Life® decreased significantly at concentrations of 100% (*p* < 0.01) and 50 % (*p* < 0.001), to (0.2 ± 0.2) × 10^9^ cells/mL and (0.112 ± 0.075) × 10^9^ cells/mL respectively (Figure 2A). In the treatment with WhiteProRoot^®^MTA and Biodentine^™^, the proliferation of cell cultures was found to be inversely proportional to the concentration, that is, the higher the concentration, the lower the cell proliferation (Figure 2A), considering the tested concentrations.

### 3.4. Cell Cycle

The cell cycle was influenced by the three biomaterials as represented in Figure 2B, with WhiteProRoot^®^MTA causing the greatest changes. There was a significant increase of the population in S-phase after treatment with WhiteProRoot^®^MTA at 50% and 100%, from 23 ± 0.7% in the control condition to 27 ± 0.4% (*p* < 0.01). Biodentine^™^ treatment at the highest concentration showed significant changes in Pre-G0 (*p* < 0.01), G0/G1 (*p* < 0.05) and G2/M (*p* < 0.001) phases.

### 3.5. Cell Death Pathways

The viabilible and apoptotic cells, late apoptotic and necrotic were not influenced by WhiteProRoot^®^MTA. Treatment with Life^®^ drastically changed cell viability and the type of cell death in a significant way and dependent on the biomaterial concentration. The Biodentine^™^ treatment negatively influenced viability at the highest concentrations with a decrease from 76.78% ± 1.44% to 54.71% ± 5.22%. Consequently, an increase of cells in apoptosis was found, from 6.57% ± 1.5% to 13.71% ± 4.49%; and an increase of cells in late apoptosis/necrosis, from 6.79% ± 0.56% to 24.14% ± 3.99%. These results can be seen in Figure 3A.

### 3.6. Mitochondrial Membrane Potential

Disruption of the mitochondrial membrane potential (MMP), which is a hallmark of apoptosis, results in decoupling of the respiratory chain and release of cytochrome C. The MMP was disrupted after treatment with conditioned media of the three biomaterials, as can be seen in Figure 3B. For treatment with Life^®^, it was found that the MMP was disturbed at the concentration of 6.25%. With WhiteProRoot^®^MTA, it was found that the MMP was significantly disturbed as it went from 2.53 ± 0.21% to 4.02 ± 0.21% (*p* < 0.001) at 50% concentration, and 4.56 ± 0.62% (*p* < 0.01) at 100% concentration. In treatment with Biodentine^™^ conditioned media at a concentration of 50%, there was a significant decrease in the monomer/aggregate ratio from 2.53 ± 0.21% to 1.69 ± 0.13% (*p* < 0.05).

### 3.7. Cellular Morphology

The cell morphology of the MDPC-23 line can be seen in Figure 3C. This cell line has a star-like central core shape. Cell morphology was maintained on exposure to conditioned media with Biodentine^™^ or WhiteProRoot^®^MTA. The images also reveal the appearance of small particles stained with dark violet which are probably sediments of the biomaterial conditioned media, and which are located between the cells, near the cytoplasmic membranes and, probably, even within the cytoplasm. There were no morphological differences in the cells between these two biomaterials. In the cell cultures treated with Life^®^, there was loss of the normal morphology and the appearance of vacuolization in the cytoplasm. The evaluation of the images also showed a modification of the cytoplasmic membrane in most cells, with the observation of cell contraction, this being the typical morphological aspect of a cell in apoptosis.

### 3.8. Oxidative Stress

Intracellular peroxide production is shown in Figure 4. In general, there was an increase of reactive oxygen species (ROS), with the White ProRoot^®^MTA and Biodentine^™^ treatment in the highest concentrations and with Life^®^ at a concentration of 6.25%.

### 3.9. Alkaline Phosphatase—Gene Expression

The expression of the ALP gene was significantly influenced by the treatments with the Life^®^, Biodentine^™^ and WhiteProRoot^®^MTA conditioned media in a concentration-dependent manner. These results can be seen in Figure 5A. The 50% concentration of Life^®^ treatment significantly decreased the ALP gene expression to 0.013 (*p* < 0.001). In cultures subjected to 6.25% Life^®^ treatment, no significant changes were observed. Treatment with WhiteProRoot^®^MTA at a concentration of 50% led to a significant decrease in gene expression with values of 0.46 relative to the control (*p* < 0.01). With Biodentine^™^, a significant increase in ALP expression was observed with values of 3.96 (*p* < 0.05) at a concentration of 6.25% and values of 8.51 (*p* < 0.05) at a 50% concentration, relative to the control.

### 3.10. Immunocytochemistry Evaluation of Sialoprotein of Dentin (DSP)

The MDPC-23 line expresses the DSP, proved as seen by marking the brownish color present in about 100% of the cell cytoplasm. In the treatment of cell cultures with a medium conditioned with Life^®^, positive labeling was observed in cells exposed to the concentration of 6.25%. In cell cultures treated with WhiteProRoot^®^MTA, the positive labeling of cells was seen for this protein. Cell cultures treated with Biodentine^™^ showed a positive DSP marking with a pattern like that of WhiteProRoot^®^MTA, but with a higher intensity of brownish. This increase in labeling intensity was observed in both concentrations of Biodentine^™^.

### 3.11. Calcium Nodules Evaluation—Morphology and Quantification

Detection of calcium nodules through the colorimetric assay with Alizarin Red S can be seen in Figure 5D. Cell images in the control condition, showed that the MDPC-23 line forms extracellular calcium deposits, as seen by the reddish-colored marking. This positive marking also occurred in cell cultures submitted to WhiteProRoot^®^MTA and Biodentine^™^. In the cell cultures exposed to treatment with Life^®^, there was a dispersed dark reddish marking.

The formation of calcium deposits by cells was influenced by the treatments with the media conditioned with Biodentine^™^ and Life^®^ in a significant manner and dependent on the concentration of the biomaterial, as compared to the control. Treatment with WhiteProRoot^®^MTA showed changes, but only between concentrations. These results, concerning the formation of calcium deposits after 120 h of exposure to the conditioned media, can be observed in Figure 5B.

## 4. Discussion

In the present study, quantitative assessments of cytotoxicity were carried out to understand some mechanisms of the reparative dentinogenesis process. The intention was to evaluate the response of the cell cultures submitted to the treatment with the biomaterials throughout the various phases of this process. In an initial phase, in which the biomaterials should stimulate the cells, the intention was to evaluate metabolic activity and cellular viability with MTT and SRB assays [23,38]. In addition to evaluating the biomaterials cytotoxic effects, studies were carried out on the types of cell death, changes in the mitochondrial membrane potential, the cell cycle, and the formation of reactive oxygen species. The next step in the process of reparative dentinogenesis is cell differentiation, when cells differentiate into odontoblast-like cells and synthesize specific proteins. It was possible to evaluate this phase with studies of cell morphology and the synthesis of two specific proteins: alkaline phosphatase and dentin sialoprotein, referred to as the major markers of odontoblast-like cells [39]. The final phase of the process, the formation of mineralized tissue, was evaluated with the Alizarin Red S assay.

The pH of a material is an essential physical property and is generally related to the pulpal response. The results showed that the pH profile of all conditioned media of biomaterials is within the alkalinity spectrum, and there are no significant differences between them. The pH of the biomaterials is related to its antimicrobial effect and ROS production. Some authors report that the increase in pH, caused by the release of calcium hydroxide, can increase ROS formation [26,40].

The response of the MDPC-23 cell line to the biomaterials was substantially different, both in the assays that assessed cell proliferation and in those that evaluated the differentiation and mineralization phases, particularly with respect to treatment with Life^®^ over the other two biomaterials. The cytotoxicity evaluation by the MTT assay is often used to obtain a screening of compounds [38]. Considering the principle of technique, MTT reduction indicates the metabolic status of cells being influenced by several factors intrinsic to cellular metabolism particularly those inherent to mitochondria [41]; therefore, in a complementary manner, the SRB assay was performed, correlating protein content with viability through quantification of cell biomass [42]. The metabolic activity and viability of MDPC-23 cells showed a marked decrease after treatment with Life^®^, irrespective of concentration. Treatment with WhiteProRoot^®^MTA did not determine changes in cellular metabolic activity and cell viability. In most cases, metabolic activity was a good indicator of cell viability.

In the case of cell cultures exposed to Biodentine^™^, maintenance of metabolic activity and viability was observed, except in the highest concentrations. In the latter case, the decrease in metabolic activity was not accompanied by such a pronounced effect in viability. Therefore, the cell cultures seem to be affected in a transient manner, but viability was not greatly compromised.

Other authors concluded that treatments with WhiteProRoot^®^MTA ensure the maintenance of cell proliferation similar to control [43,44], whereas in the treatments with Life^®^ there was a decrease [45,46], particularly at the highest concentrations.

In the sequence of apoptosis, if the damage is very high, cell death by necrosis may occur. In our study with Life^®^ and Biodentine^™^ at high concentrations, it was observed that the decrease in viability occurred by apoptosis, rather than necrosis. This may occur because, at an early stage, the damage caused by cytotoxic agents occurs at the cytoplasmic level, since the nuclear membranes of these cells remain intact [40]. Apoptosis does not occur during the cell differentiation phase, due to increased expression of proteins involved in the inhibition of calcium phosphate complex formation [39].

The MMP reflects the electron transport chain activity as well as the mitochondrial function. Disruption of MMP is a characteristic of apoptosis [26]. MMP disruption occurred after treatment with the highest concentrations of WhiteProRoot^®^MTA. Thus, mitochondrial function was affected, however, without consequent loss of cell viability. MMP was slightly decreased after treatment with Biodentine^™^, suggesting that cell death by late apoptosis and necrosis is independent of the mitochondrial pathways.

The increase in the number of cells in S and G2/M phases observed after treatment with WhiteProRoot^®^MTA and Biodentine^™^ was also reported in other studies with MTA, suggesting an increase of cells in the proliferation phase and a potentially positive effect in the regeneration of pulp tissue in vivo [47].

The intracellular production of peroxides indicates the redox state of the cells and allows understanding the cellular responses to biomaterials. Reactive oxygen species are a natural byproduct of oxygen metabolism and play a key role in regulating other cellular signaling, proliferation and survival pathways. On the other hand, an excessive increase in the production of peroxides during cell stress results in significant damage to the cell structure, as is typical of neurodegenerative diseases, cancer or aging [40]. In fact, an increase in peroxide production was observed after treatment with WhiteProRoot^®^MTA and with Biodentine^™^. However, the increases observed in this study may not surpass the threshold of deleterious effects considering MDPC-23 cells [48]. In fact, the production of ROS observed might be beneficial and important for cell differentiation during repair dentinogenesis and mineralized matrix formation. The slight increase in ROS that occurred with all the biomaterials may have repercussions on its antimicrobial effect.

In the treatments with the biomaterials WhiteProRoot^®^MTA and Biodentine^™^, we observed a considerable increase in the alkaline phosphatase, through expression of the alkaline phosphatase gene, in the synthesis of dentin sialoprotein and in the formation of calcium nodules. These increases in the production of specific proteins of the differentiation phase and the formation of calcium nodules are correlated with each other, as observed in other studies [49]. This was observed in the higher concentrations, being more evident in the treatments with Biodentine^™^.

The cytotoxicity of the treatment with Biodentine^™^ and WhiteProRoot^®^MTA biomaterials, varies from moderate to weak, being much lower than the 40 to 45% values accepted in the ISO standard. With the Life^®^ treatment, this acceptable level of cytotoxicity was observed only at the lowest concentration. These levels of toxicity resemble what appears to occur in vivo in pulp tissue when these materials are used for direct pulpal capping. In vivo, the outer cellular layers in contact with the protective material act, presumably, as a filter or barrier, protecting the inner layers. This behavior associated with pulp clearance necessarily decreases the magnitude of cytotoxicity. This explains why, despite the caustic nature of some of these materials, their efficacy and clinical success is demonstrated [50]. The significant differences demonstrated between the calcium hydroxide-based cement and the mineral trioxide aggregates and tricalcium silicate cements are attributed to the structural differences in their basic components and consequently to different biochemical reactions generated. It is established that the mineral trioxide aggregates induce the formation of a hard tissue barrier thicker than the calcium hydroxide-based materials, accompanied by less inflammation of the adjacent tissues [51]. In addition, MTA increases the differentiation of progenitor cells into odontoblasts, by the induction of secretion of morphogenetic proteins and growth factors such as BMP-2 and TGF-β1 [11,46].

Biodentine^™^ reacts through the hydration of the tricalcium silicate and the production of a calcium silicate-based calcium hydroxide gel that in contact with the phosphate ions has the capacity to precipitate a hydroxyapatite-like compound [20]. The repair process occurs by the deposition of the mineralized tissue and its stimulation depends on the pH and on the ability of the tricalcium silicate cement to release various ions. Extracellular calcium ions are known to be involved in proliferation and differentiation signaling pathways. The calcium ions released from the tricalcium silicate cements activate the MAPK signaling pathway, thus modelling odontoblastic differentiation and mineralization genes [20,52,53]. This was corroborated in our study, with an increase in RNA alkaline phosphatase gene expression, increase in sialoprotein synthesis, and in the formation of calcium deposits [54].

Calcium hydroxide-based cements decreased metabolic activity and cellular viability, with a marked increase in cell death, considerable changes in the cell cycle and no protein synthesis or calcium nodules formation. Therefore, calcium hydroxide-based cements negatively influence the proliferation of pulp cells in vitro. This material leads to tissue superficial necrosis and apoptosis [2]. The local inflammatory process stimulates dental pulp stem cells to migrate to the lesion site and differentiate into odontoblasts, which will synthesize precursor mineralization proteins and later form mineralized deposits [6,8]. These evidences justify this material use during decades with a relatively high success.

Concerning the mineral trioxide aggregates and tricalcium silicates materials, we observed increased metabolic activity and cell viability, with a high percentage of living cells and without interference in the cell cycle. Also, in the later stages of differentiation and mineralization, the tricalcium silicate cements had a better performance [55], with a marked increase of alkaline phosphatase expression, of dentin sialoprotein and calcium nodule formation than the mineral trioxide aggregate cements These results, together with the ability of these materials to promote a moderate inflammatory response [50], might contribute to the higher clinical success and less tunnel defects [17,18,50].

Hence, this work supports mineral trioxide aggregate-based cements indication for pulp capping therapeutics currently recognized as the gold standard. However, tricalcium silicate-based cements may be a promising alternative, since they promote the same cellular response as MTA. Additionally, this material surpasses some of its important disadvantages, namely the difficulty of clinical manipulation, the high setting time which may be incompatible with the clinical times, the high cost, and the potential of discoloration of the dental structure [19].

## 5. Conclusions

Calcium hydroxide-based cements decreased metabolic activity and cellular viability, with a marked increase in cell death, considerable changes in the cell cycle, and no protein synthesis or calcium nodules formation.Mineral trioxide aggregates and tricalcium silicates materials increased metabolic activity and cell viability, with a high percentage of living cells and without interference in the cell cycle. Also, in the later stages of differentiation and mineralization, the tricalcium silicate cements had a better performance, with a marked increase of alkaline phosphatase expression, of dentin sialoprotein and calcium nodule formation compared to the mineral trioxide aggregate cements.This work supports tricalcium silicate, and mineral aggregate trioxide cements-based can be indicated for pulp capping therapeutics.

## Figures and Tables

**Figure 1 materials-12-01184-f001:**
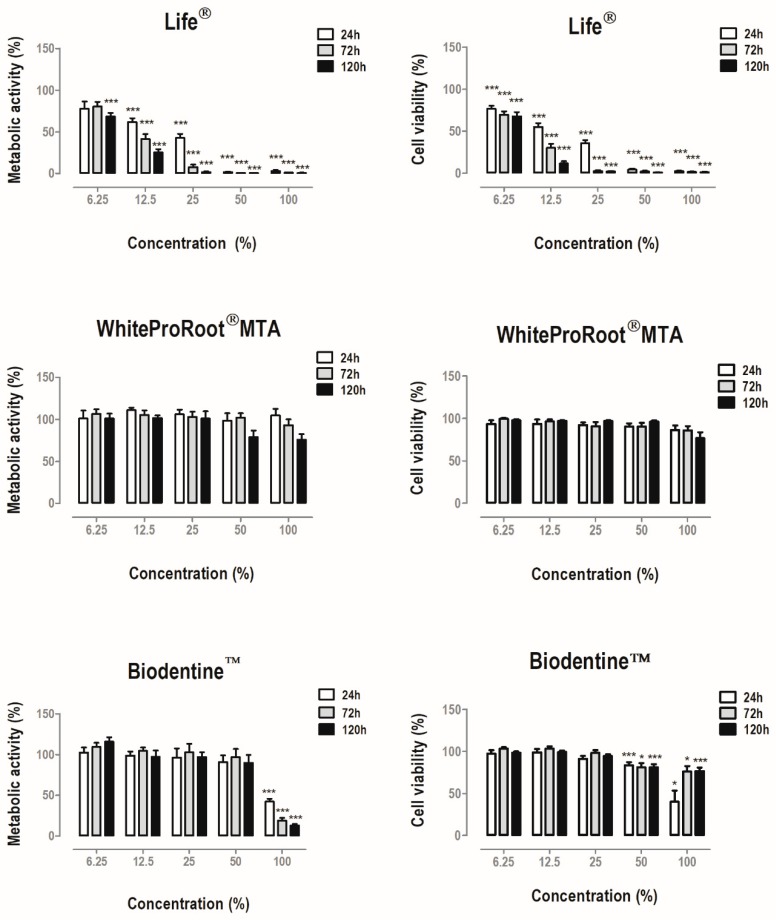
Metabolic activity and cell viability of MDPC-23 cells treated with Life^®^, WhiteProRoot^®^MTA and Biodentine^™^ conditioned media for 24, 72 and 120 h. The results represent the mean and standard error of six independent duplicate assays. Significant differences are represented by *, where * means *p* < 0.05, ** means *p* < 0.01 and *** means *p* < 0.001.

**Figure 2 materials-12-01184-f002:**
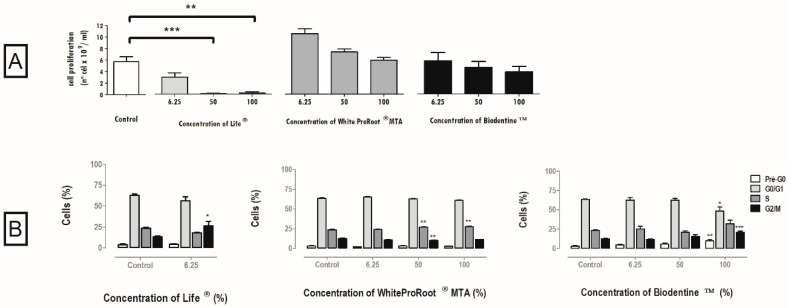
(**A**) Cell proliferation after 120 h of incubation with Life^®^, WhiteProRoot^®^MTA and Biodentine^™^ conditioned media. All the results express the mean and standard error of at least seven independent assays. (**B**) Cell cycle in MDPC-23 cells treated with Life^®^, WhiteProRoot^®^MTA and Biodentine^™^ conditioned media after 120 h of exposure. The results are represented as percentage of cells in the pre-G0, G0/G1, S and G2/M phases. The results represent the mean and standard error of five trials. Significant differences from control or between conditions are represented by *, where * means *p* < 0.05, ** means *p* < 0.01 and *** means *p* < 0.001.

**Figure 3 materials-12-01184-f003:**
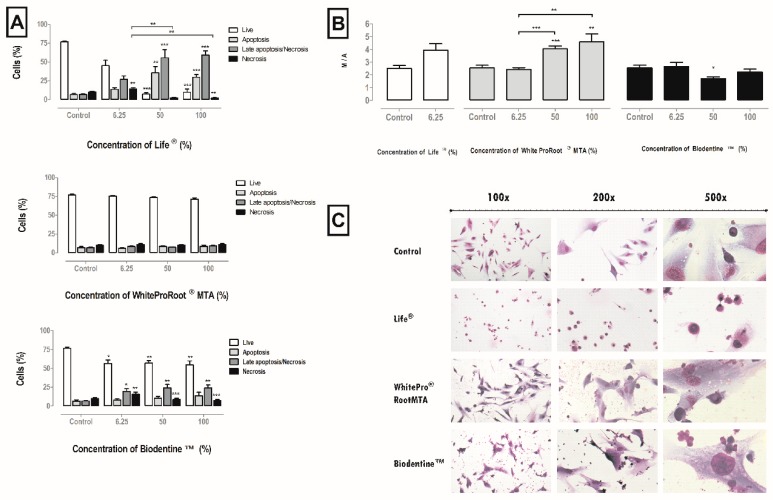
(**A**) Cell viability and types of death in MDPC-23 cells treated with Life^®^, WhiteProRoot^®^MTA and Biodentine^™^ conditioned media after 120 h of exposure. Results represent the percentage of living cells, cell in apoptosis, in late apoptosis or necrosis, and in necrosis. The results represent the mean and standard error of six trials. (**B**) MMP of MDPC-23 cells treated with Life^®^, WhiteProRoot^®^MTA and Biodentine^™^ after 120 h of exposure. The results represent the ratio monomers/aggregates (M/A) for each condition. The results represent the mean and standard error of five trials. In all results, significant differences from control or between conditions are represented by *, where * means *p* < 0.05, ** means *p* < 0.01 and *** means *p* < 0.001. (**C**) MDPC-23 cells stained with May-Grünwald Giemsa after treatment with 50% concentration of biomaterials conditioned medium. The control group represents cells in culture in DMEM with 10% FBS. Images in the left column were obtained with a magnification of 100×; the images in the central column were obtained with a magnification of 200×; and the images in the column on the right were obtained with a magnification 500×. These assays were performed in duplicate.

**Figure 4 materials-12-01184-f004:**
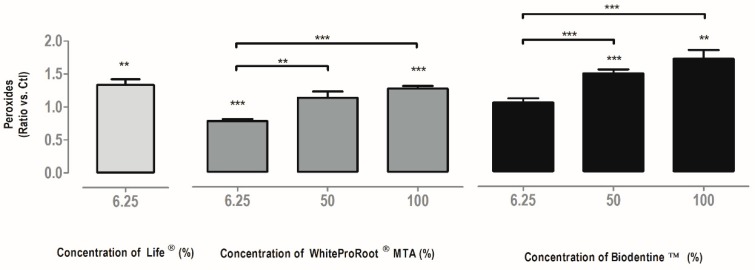
Intracellular peroxide production in MDPC-23 cells treated with Life^®^, WhiteProRoot^®^MTA and Biodentine^™^ after 120 h of incubation. The results are presented as a variation against control cell cultures. All the results express the mean and standard error of at least seven independent assays. In all results, significant differences from control or between conditions are represented by *, where * means *p* < 0.05, ** means *p* < 0.01 and *** means *p* < 0.001.

**Figure 5 materials-12-01184-f005:**
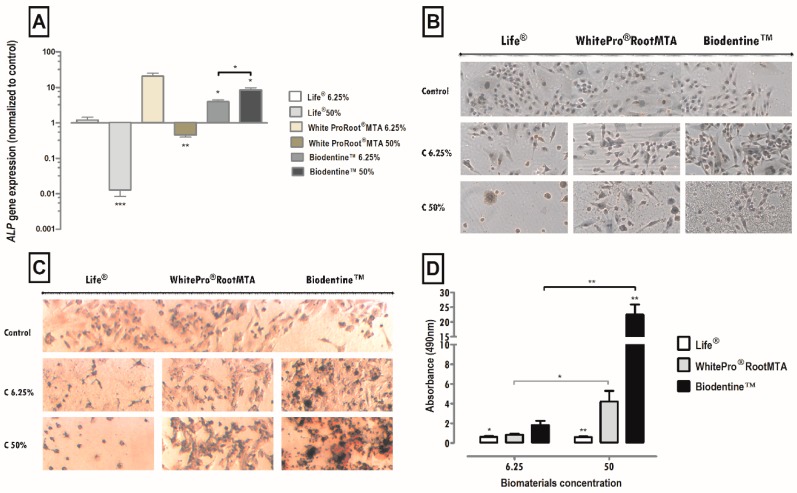
(**A**) Alkaline phosphatase gene expression by MDPC-23 cells treated with Life^®^, WhiteProRoot^®^MTA and Biodentine^™^ after 96 h of exposure. The results represent the mean and standard error of four trials. (**B**) Images from cultured MDPC-23 cell lines labeled by immunocytochemistry for the detection of DSP expression when subjected to treatment with Life^®^, WhiteProRoot^®^MTA and Biodentine^™^ at concentrations of 50% and 6.25% after 96 h of incubation. (**C**) Images from cultured MDPC-23 cells stained with Alizarin Red S stain when treated with Life^®^, White ProRoot^®^MTA and Biodentine^™^ biomaterials at concentrations of 50% and 6.25% after 120 h of incubation. All the photographs were obtained with a magnification of 100×. These assays were performed in duplicate. (**D**) Formation of calcium deposits from MDPC-23 cells treated with Life^®^, WhiteProRoot^®^MTA and Biodentine^™^ after 120 h of exposure. The results represent the mean and standard error of six trials and are the relationship between the absorbances obtained between the study condition and the control. For (**A**,**D**), significant differences are represented by *, where * means *p* < 0.05, ** means *p* < 0.01 and *** means *p* < 0.001.

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
