# Peer review of "Biodentine Boosts, WhiteProRoot®MTA Increases and Life® Suppresses Odontoblast Activity"

_materials, 2019, doi:10.3390/ma12071184_

Reviewer 1 Report

The study evaluated biological effect of 3 different dental pulp capping materials; WhiteProRoot MTA, Biodentine and Life, on odontoblast-like mouse cell line (MDPC-23). The study is well designed and executed with plenty of data, but there are some concerns.

Title of the manuscript seems too definitive. Author used odontoblast-like mouse cell line (MDPC-23) for this study, and all of studies are in vitro based. There has been no ‘pulp capping’ study such as the method outlined is International Standard, ISO 7405. Also, in ‘2. Materials and Methods’ under ‘Cell Culture’, information on MDPC-23 cells are lacking and wondering if process of immortalization or cell culturing would have influenced the nature of cells. Hence, it may be ideal to review methods and also the title.

Some of experiments are carried out using dilutions of extract from the biomaterials. What would be the basis of choosing 6.25% and 50%? How would authors be certain that exposure dose would be same for all three materials?

In abstract, authors stated that the aim of the study ‘was to assess the cytotoxicity of three different direct pulp capping materials’. This has been also stated in ‘1. Introduction’ at the end. Cytotoxicity is an indication of toxicity towards cells, which would indicate biocompatibility of the material. However, authors also seem to have focused on biological effect beyond biocompatibility; bioactivity. What would be the basis/background on studying bioactivity for three materials? It would be ideal to revise abstract and introduction.

Author Response

Response to Reviewer 1

The study evaluated biological effect of 3 different dental pulp capping materials; WhiteProRoot MTA, Biodentine and Life, on odontoblast-like mouse cell line (MDPC-23). The study is well designed and executed with plenty of data, but there are some concerns.

Title of the manuscript seems too definitive. Author used odontoblast-like mouse cell line (MDPC-23) for this study, and all of studies are in vitro based. There has been no ‘pulp capping’ study such as the method outlined is International Standard, ISO 7405.

R: Title was changed in the manuscript.

Also, in ‘2. Materials and Methods’ under ‘Cell Culture’, information on MDPC-23 cells are lacking and wondering if process of immortalization or cell culturing would have influenced the nature of cells. Hence, it may be ideal to review methods and also the title.

R: The MDCPC-23 cell line was spontaneously immortalized as clarified in the manuscript. Moreover, these cells continue to show characteristics of odontoblasts cells that we could confirm while collecting the data for this paper, namely, the expression of DSP and ALP and the production of mineralized aggregates. This features evidence that MDPC-23 retain their original phenotype.

Some of experiments are carried out using dilutions of extract from the biomaterials. What would be the basis of choosing 6.25% and 50%?

R: We started this research by evaluating metabolic activity and cell viability. For this we considered the conditioned media (100%) and further dilutions (50%, 25%, 12.5%, 6.25%). The rational for this was based in the instructions from ISO standard 10993-5. We verified that mostly there were no significant differences between 6.25%, 12.5% and 25%. With this we explored type of cells death, cell cycle, reactive oxygen species, mitochondrial membrane potential considering 6.25, 50% and 100%. In the same manner we verified that 50% and 100% concentrations were the most deleterious by opposition to 6.25%. Therefore, we finalized studying the concentrations 6.25% and 50%.

How would authors be certain that exposure dose would be same for all three materials?

R: The concentrations were obtained by dilution of a conditioned media. This media was prepared as described in the ISO standard 10993-5 that states the contact of a determined volume of media with a certain area of the materials during 24 hours at 37º.

In abstract, authors stated that the aim of the study ‘was to assess the cytotoxicity of three different direct pulp capping materials’. This has been also stated in ‘1. Introduction’ at the end. Cytotoxicity is an indication of toxicity towards cells, which would indicate biocompatibility of the material. However, authors also seem to have focused on biological effect beyond biocompatibility; bioactivity. What would be the basis/background on studying bioactivity for three materials? It would be ideal to revise abstract and introduction.

R: Alterations were made in the manuscript. Moreover, we recognized the three materials have a similar chemical composition while clinical behavior is significantly different. (In fact, we also have an in vivo study submitted for publication on this and a systematic review published with this conclusion. - doi: 10.1016/j.jebdp.2018.02.002) Therefore, we considered important to evaluate the effect of the three compounds at the cellular level. 

Reviewer 2 Report

Authors described the effect of media conditionated by eluting 3 commercial materials (Biodentine, WhiteProRootMTA and Life) on osteoblasts MDPC-23.

Authors studied viability, proliferation, cell cycle, oxidative stress, apoptosis etc.

The paper is clear and the experiments technically well performed. However, some figures (and data) do not show control that should be the unconditioned media (figg 1, 4, 5)

Even from graphical point of view it is hard to read correctly the histograms. I suggest to widen them and to avoid “zebra” style filling.

Author Response

Response to Reviewer 2

Authors described the effect of media conditionated by eluting 3 commercial materials (Biodentine, WhiteProRootMTA and Life) on osteoblasts MDPC-23.

Authors studied viability, proliferation, cell cycle, oxidative stress, apoptosis etc.

The paper is clear and the experiments technically well performed. However, some figures (and data) do not show control that should be the unconditioned media (figg 1, 4, 5)

R: The control is not showed because results are presented as a normalization of control experiences.

Even from graphical point of view it is hard to read correctly the histograms. I suggest to widen them and to avoid “zebra” style filling.

R: Alterations were made in the manuscript.

Round  2

Reviewer 2 Report

Yes. You agree

Author Response

Thank you for the reply.